

# Floquet engineering of axion and high-Chern number phases in a topological insulator under illumination

Mohammad Shafiei[1,2], Farhad Fazileh[2],
François M. Peeters[1,3] and Milorad V. Milošević[1,4*]

**1** Department of Physics, University of Antwerp,
Groenenborgerlaan 171, B-2020 Antwerp, Belgium
**2** Department of Physics, Isfahan University of Technology, Isfahan 84156-83111, Iran
**3** Departamento de Fisica, Universidade Federal do Ceará, 60455-760 Fortaleza, Ceara, Brazil
**4** NANOlab Center of Excellence, University of Antwerp, 2020 Antwerp, Belgium

⋆ milorad.milosevic@uantwerpen.be

## Abstract

Quantum anomalous Hall, high-Chern number, and axion phases in topological insulators are characterized by its Chern invariant $C$ (respectively, $C = 1$, integer $C > 1$, and $C = 0$ with half-quantized Hall conductance of opposite signs on top and bottom surfaces). They are of recent interest because of novel fundamental physics and prospective applications, but identifying and controlling these phases has been challenging in practice. Here we show that these states can be created and switched between in thin films of $Bi_2Se_3$ by Floquet engineering, using irradiation by circularly polarized light. We present the calculated phase diagrams of encountered topological phases in $Bi_2Se_3$, as a function of wavelength and amplitude of light, as well as sample thickness, after properly taking into account the penetration depth of light and the variation of the gap in the surface states. These findings open pathways towards energy-efficient optoelectronics, advanced sensing, quantum information processing and metrology.

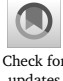

# 1 Introduction

Topological insulators (TIs) attracted strong interest over the last decade due to their unique fundamental properties and numerous envisaged applications, especially in spintronics and quantum computing [1, 2].

Due to strong spin-orbit coupling, incorporating magnetization to the $Bi_2Se_3$ family of materials - being the most well-known three-dimensional TIs - can lead to otherwise unattainable magnetic topological states, such as the quantum anomalous Hall (QAH) state - also known as Chern insulator [3, 4], the high-Chern number QAH states [5], and axion insulator (AI) phase [6–8]. One should note however that QAH states can be observed in some structures where spin-orbit coupling is weak, like multilayer graphene [9]. Magnetization in a TI is typically achieved via applied magnetic field, doping with magnetic atoms, or heterostructuring with magnetic adlayers. Depending on the magnitude and direction of induced magnetization, QAH, high-Chern number or AI states can be stabilized and investigated further [10–14].

In the absence of an external magnetic field, QAH effect can be realized if the directions of applied magnetization on the top and bottom surfaces are the same. The potential applications of the QAH effect in spintronic devices [15], quantum metrology [16], and Majorana edge modes [17] have garnered significant attention in recent years. These states may also be used as an electrical resistance standard since they can exist without or in a weak magnetic field, unlike the quantum Hall effect that requires a rather strong magnetic field. Increasing the size of magnetization such that the induced magnetization gap is larger than the bulk gap of the system leads to the emergence of the high-Chern number QAH phase [11], where integer Chern invariant $C > 1$ gives the number of open conducting chiral edge channels [18, 19], and yields correspondingly higher conductance compared to QAH (where $C = 1$), namely $(\sigma_{xx}, \sigma_{xy}) = (0, Ce^2/h)$. Therefore, facilitated the realization of high-Chern number insulators provides opportunities for multichannel quantum computing [20], higher-capacity circuit interconnects, and energy-efficient electronic devices [21].

The axion insulator state can appear in magnetic topological insulators once the direction of magnetization is opposite on opposite surfaces. AI phase is characterized by half-quantized Hall conductance on opposite surfaces, and a Chern number $C = 0$. Besides opening pathways to the production of non-reciprocal thermal emitters [22] and the detection of Majorana fermions [23], the realization of these states in TIs is also valuable as an accessible condensed-matter analogue of axion particles in high energy physics [13, 24, 25].

Many attempts have therefore been made to stabilize these novel magnetic topological states in recent years. However, identifying them unambiguously in experiment has been challenging. For example, realizing high-Chern number QAH states requires an exchange field larger than the bulk band gap, which is prohibited in practice by the detrimental effects of the high doping levels needed [26]. These requirements can be softened by applying in-plane strain [11], which has been proven difficult experimentally for the $Bi_2Se_3$ family of topological insulators.[1] Note that high-Chern states have also been identified in multilayer systems [27], but there the high Chern number stems from the surface states of separate layers that are added together. In our system, the states with a high-Chern number are related to bulk topology and additional chiral edge channels forming due to bulk states [11, 26]. On the other hand, AI state can be stabilized in TI/magnetic heterostructures [10, 18], but magnetic layers on the top and bottom of the TI must have opposite magnetization orientation - which is nontrivial in practice.

---

[1]As these are layered van der Waals materials, when thin films are grown on a substrate with a mismatched lattice to induce strain, the lattice constant in the layers adjacent to the interface immediately relaxes to the bulk value, and there is effectively no net observable strain.

In this article, we abandon the standard techniques such as strain and doping, and explore tuning of the topological phases of a TI using Floquet engineering [28], specifically we introduce circularly-polarized light (CPL) as a time-periodic electromagnetic perturbation. Illumination is a convenient external tool to control the properties and phases of material, which has the important advantage over conventional mechanical and electrical techniques that it does not require direct contact with the material. It is also ultra-fast [29,30], in principle non-destructive, and more versatile (as phase transitions can be induced/probed by both intensity and wavelength of the light). Recent advances in laser and spectroscopic technology have reignited interest in Floquet engineering as an approach to induce and control the fascinating quantum properties [28,29,31]. Once TI is exposed to circularly-polarized light, time-reversal symmetry (TRS) breaks, and a gap in the surface states appear. Here the chirality of the light determines the sign and direction of the induced chiral current [29]. This yields confidence that using light to create axion states or Chern insulator phases could mitigate the previously encountered practical obstacles. It has been suggested that CPL could create topological phase transitions in antiferromagnetic topological insulator $MnBi_2Te_4$ films [32]. However, QAH states with high Chern numbers can only be achieved by modulating the van der Waals gap in the surface layers. Also, recent research has shown that in some cases the filling of Floquet bands does not conform to the Fermi-Dirac statistics. As a result, the so-called Floquet topological states may not lead to quantized transport [33]. On the other hand, It has been demonstrated that two-dimensional materials' opacity to CPL across a broad frequency range can generate a spectral function that integrates over frequencies to yield the Chern number and high Chern number QAH state [34]. This optical experiment thus offers a straightforward approach to measuring the Chern number [35]. In this work, we avoid such practical challenges and predict the topological phases in standard $Bi_2Se_3$ films, as a function of the sample thickness and parameters of the illumination, to reveal that the axion state and high-Chern number phases can indeed be induced and precisely controlled using ultraviolet light.

## 2 Tight-binding methodology with Floquet theory

In this article, we consider $Bi_2Se_3$ as the prototypical three-dimensional TI, that has been widely investigated due to its chemical stability, ease of synthesis, and simple band structure [36,37]. $Bi_2Se_3$ has a rhombohedral crystal structure with $R\bar{3}m$ symmetry, whereby the atoms are arranged in a triangular lattice with ABCABC stacking, with a quintuple layer (QL) as the unit cell [37, 38]. We consider a $Bi_2Se_3$ film of thickness expressed as a number of quintuple layers $N_{QL}$, and employ a real-space tight-binding Hamiltonian as follows [39,40]:

$$H = \Sigma_i c_i^\dagger E_{on} c_i + \Sigma_{i,\alpha}(c_i^\dagger T_\alpha c_{i+\alpha} + H.c),\tag{1}$$

where $\alpha=1,2,3,4$, and the operator $c_i^\dagger$ ($c_i$) creates (annihilates) an electron at site $i$. Further we have:

$$E_{on} = (E_0 - 2\Sigma_\alpha B_\alpha)\sigma_z \otimes \sigma_0,\tag{2}$$

and

$$T_\alpha = C_\alpha \sigma_0 \otimes \sigma_0 + B_\alpha \sigma_z \otimes \sigma_0 - i\left(\frac{A_\alpha}{2}\right)\sigma_x \otimes \sigma \cdot \mathbf{n}_\alpha.\tag{3}$$

Considering the nearest neighbors only, each unit cell will have 6 neighboring unit cells in the x-y plane that are connected by vectors $\mathbf{n_i}$ (i=1,2,3) and two unit cells in the $z$ direction connected by vector $\mathbf{n_4}$, where $\mathbf{n_1}=(1/2,\sqrt{3}/2,0)$, $\mathbf{n_2}=(-1/2,\sqrt{3}/2,0)$, $\mathbf{n_3}=(1,0,0)$, and $\mathbf{n_4}=(0,0,1)$. The hopping parameters ($A_\alpha, B_\alpha, C_\alpha$) are related to the effective four-band Hamiltonian near the $\Gamma$ point, as detailed in Ref. [10]. By fitting the band structure of this Hamiltonian with DFT data and considering particle-hole symmetry in the system without losing generality ($C_\alpha=0$), we consider the parameters of this Hamiltonian as $A_{x,y} = 0.5$ eV, $A_z = 0.44$ eV, $E_0 = 0.28$ eV, $B_{x,y} = B_z = 0.25$ eV.

Applying monochromatic light with frequency $\omega$ to the system causes light to interact with electrons, and the Hamiltonian of the system becomes $H(t) = H(t + 2\pi/\omega)$. The system may be described using the Floquet theorem [41], where illumination is considered as a time-periodic perturbation. The primary objective of the Floquet method is to extend the quantities in Fourier modes $\exp(-im\omega t)$, where $m$ $(= 0, \pm 1, ...)$ is a lattice site index in a fictitious Floquet irradiation direction and $\omega$ is the driven frequency [42]. Similar to the Bloch theorem that describes spatial periodicity in crystals induced by translational symmetry, the Floquet time-crystal theory describes temporal periodicity [42, 43]. The interaction between the system at time $t$ and its temporal images at time $t + nT$ is analogous to the interaction between an atom and its spatial image in the neighboring unit cells. We consider the weak periodic perturbation and the frequency $\omega$ larger than the system's energy scale, avoiding resonant transition. In this situation, the system may be described by a time-independent Hamiltonian with high-frequency expansion. In the high-frequency regime, expansions such as van Vleck [44], Floquet-Magnus [45], or Brillouin-Wigner [46], are typically used to describe the system in terms of a time-independent Hamiltonian.

According to Peierls substitution, the time-dependent tight-binding Hamiltonian $H$ in the presence of a periodic field $\mathbf{A}$ becomes as follows [29]:

$$\langle m_0|\tilde{H}(t)|n_R\rangle = \langle m_0|H|n_R\rangle \, e^{ie/\hbar \mathbf{A}\cdot(\mathbf{R}+\tau_n-\tau_m)}, \tag{4}$$

where $\tau_n$ is the center of the $|n_0\rangle$ orbital. Bloch waves are made based on atomic orbitals as follows:

$$|n_k\rangle = \frac{1}{\sqrt{N}} \sum_R e^{i\mathbf{k}\cdot(\mathbf{R}+\tau_n)} |n_R\rangle, \tag{5}$$

so the time-dependent Hamiltonian in the basis of Bloch waves becomes:

$$\tilde{H}(k,t) = \langle m_k|\tilde{H}(t)|n_k\rangle = H\left(k + \frac{e}{\hbar}A(t)\right). \tag{6}$$

For field $A$ with time period $T$, Fourier transformation of $\tilde{H}(k,t)$ is equal to:

$$\tilde{H}^m(k) = \frac{1}{T} \int_0^T e^{-im\omega t} H\left(k + \frac{e}{\hbar}A(t)\right) dt, \tag{7}$$

therefore, the effective Floquet Hamiltonian in the high-frequency approximation is as follows [29, 42, 47]:

$$H_{eff}^F = \tilde{H}^0 + \sum_{m\neq 0} \frac{1}{2m\hbar\omega}\left[\tilde{H}^{-m}, \tilde{H}^m\right], \tag{8}$$

where $\omega = 2\pi/T$ is the angular frequency, and we have applied the van Vleck expansion [29, 48].

# 3 Fundamentals of thin-film topological insulators under illumination

We first investigate the effect of applying CPL to the top surface of the TI. The application of CPL causes TRS to collapse, and the addition of the mass term to the Hamiltonian confirms the emergence of nontrivial TI states via Berry curvature. As mentioned, Floquet states can be used to express the evolution of the system in terms of effective and static Hamiltonian if the system is in the off-resonant regime - i.e. the driving frequency $\omega$ is larger than all energy scales in the system, such as the gap.
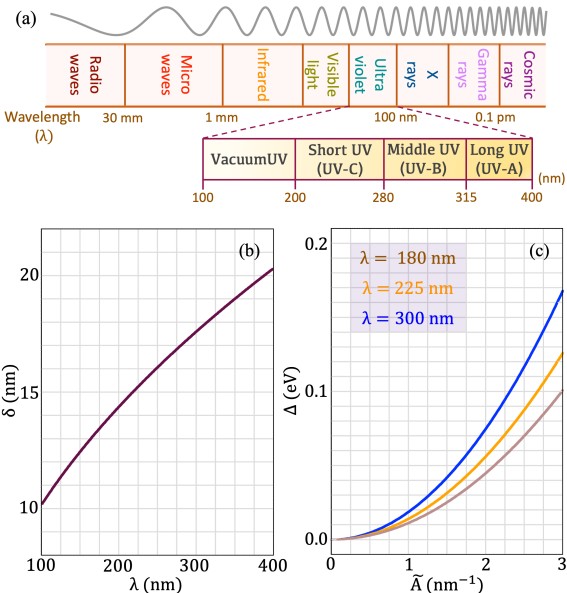

Figure 1: (a) The electromagnetic spectrum as a function of the wavelength. (b) Penetration depth in terms of ultraviolet wavelength for light of intensity $\tilde{A}$=2.5 nm$^{-1}$. (c) The gap of the surface state ($\Delta$) for applied CPL of wavelength $\lambda = 180$, 225, or 300 nm, as a function of the light intensity $\tilde{A}$.

The electromagnetic spectrum from radio waves to cosmic rays is sketched in Fig. 1(a). In what follows, we consider ultraviolet light to manipulate the topological phases in Bi$_2$Se$_3$, with wavelengths from 100 to 400 nm, which is considered a high-frequency range. Increasing the radiation frequency beyond this range would cause the edge state in Bi$_2$Se$_3$ to shift from a dissipationless quantized charge transport behavior to a dissipative domain without quantization, destroying the system's topological behavior [41].

In addition to frequency limitations, it should be noted that the intensity of the applied light cannot exceed a certain threshold, as doing so it can disrupt the TI structure. On the other hand, low-intensity illumination causes a small gap, making it more challenging to identify magnetic topological states in practice. Considering the optical field parameters in the high-frequency regime, the maximum intensity of light that can be applied is $1/a\sqrt{\hbar\omega/B}$, where $a$ and $B$ are lattice constant and the corresponding hopping parameter of the Hamiltonian, respectively. According to the parameters of Eq. (1), the maximum permitted light intensity for Bi$_2$Se$_3$ is approximately [49,50] $\tilde{A} \approx 4$ nm$^{-1}$. CPL vector potential is expressed as $\mathbf{A} = A_0(\cos\omega t, \eta\sin\omega t, 0)$, where $\eta = \pm 1$ denotes the right/left chirality of circularly polarized light. For convenience, in what follows the used intensity of light $A_0$ will be given in reduced form as $\tilde{A} = eA_0/\hbar$.

When irradiating the surface of a TI with light, the penetration depth will depend on the resistivity of the bulk material ($\rho$), the wavelength of the incident light ($\lambda$), as well as the permeability ($\mu$) and permittivity ($\epsilon$) of the material. The penetration depth based on these parameters can be expressed as [51]

$$\delta = \sqrt{\frac{\lambda\rho}{\pi c\mu}}\sqrt{\sqrt{1+\left(\frac{2\pi c\rho\epsilon}{\lambda}\right)^2}+\frac{2\pi c\rho\epsilon}{\lambda}}, \qquad (9)$$

where $c$ is the speed of light, and the bulk resistance of Bi$_2$Se$_3$ is $1.136 \times 10^{-5}$ m/S [52,53]. Fig. 1(b) shows the calculated penetration depth $\delta$ as a function of the ultraviolet wavelength

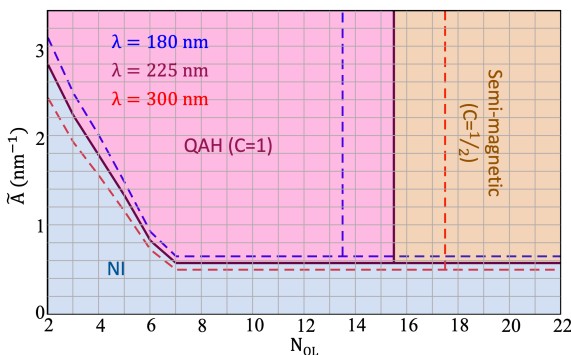

Figure 2: The topological phase diagram of thin-film Bi$_2$Se$_3$ with top surface exposed to light (down to the calculated penetration depth, cf. Fig. 1(b)), including normal-insulator (NI), QAH, and semi-magnetic states, as a function of applied light intensity $\tilde{A}$ and sample thickness N$_{QL}$, for wavelengths $\lambda = 180$, 225, and 300 nm.

(i.e. the intensity of applied light decays exponentially inside the material, following the Beer-Lambert law [54]). We consider the intensity of light in the model as $\tilde{A}\exp(-N_z/\delta)/(N_z + 1)$, where $N_z$ is the distance from the irradiated plane in the vertical direction in terms of quintuple layers. For instance, light with wavelength $\lambda = 200$ nm and intensity $\tilde{A} = 2.5$ nm$^{-1}$ has a penetration depth of approximately $\delta = 14.5$ nm, corresponding to a thickness of 15 QL. The magnitude of the gap triggered by applied CPL also depends on the intensity and wavelength of light. In Fig. 1(c), the calculated gap of the surface state is shown as a function of $\tilde{A}$ for light of wavelength $\lambda = 180$, 225, or 300 nm, which correspond to energies $\hbar\omega = 5$, 4, and 3 eV, respectively. As shown in Fig. 1(c), the surface states gap can be relatively broadly tuned by varying the wavelength for a given light intensity.

Upon above prime considerations, we investigate the effect of applied CPL only on one surface of the TI, and the incurred phase transitions as a function of the light intensity ($\tilde{A}$) and the sample thickness (N$_{QL}$). Once the sample thickness is sufficiently small for the surface wave functions to overlap, a hybridization gap will emerge [36]. Consequently, the hybridization gap decreases as sample thickness is increased; for 2-6 QL samples in our model, the found hybridization gap was 110, 70, 45, 25, and 10 meV, respectively [19]. In these cases, the system will transit from the normal insulator (NI) phase to the QAH phase with $C = 1$ when the intensity of the applied light is increased, as soon as the gap created by the light exceeds the hybridization gap. For given ultrathin thickness of the sample, for larger wavelength of the light a smaller amplitude is required to induce the transition, following the gap dependence shown in Fig. 1(c). Moreover, the penetration depth of light changes when modifying the wavelength. As the wavelength is increased, the penetration depth of light increases as well, and the transition from QAH to semi-magnetic phase consequently shifts to larger thickness of the sample.

If the sample thickness exceeds the hybridization range, the hybridization gap vanishes, and weak applied light readily induces the phase transition from NI to QAH, provided that the sample thickness is smaller than the penetration depth of the light. When the sample thickness is larger than the penetration depth of the applied light, only the exposed surface state is gapped, while the opposite surface state remains gapless. In this case, the Chern invariant will be $C = 1/2$ on the exposed surface and $C = 0$ on the opposite surface, and the system will be in the semi-magnetic phase with a Chern number of $C = 1/2 + 0 = 1/2$. Fig. 2 shows our calculated topological phase diagram of NI, QAH and semi-magnetic states as a function of the light intensity $\tilde{A}$ and the sample thickness N$_{QL} = 2-22$, driven by light of wavelength $\lambda = 180$, 225, or 300 nm. In the calculation of the Chern number, the effect of illumination is added

to the Hamiltonian's on-site energy, taking into account the layer index in the sample. To do this, we first calculate the depth of light penetration based on its characteristics - intensity and frequency. Once we have determined the penetration depth, we add the effect of light to the Hamiltonian, exponentially decaying in the real space based on the layer number. The Chern number $C$ for a particular band can then be expressed as an integral over the Brillouin zone [8, 55]:

$$C = \frac{1}{2\pi} \int_{BZ} \Omega(k) \, dk, \tag{10}$$

where $\Omega(k)$ is the Berry curvature as:

$$\Omega(k) = \nabla_k \times A(k). \tag{11}$$

$A(k)$ stands for the Berry connection:

$$A(k) = -i \sum_{n \neq m} \frac{\langle u_{nk} | \nabla_k | u_{mk} \rangle \langle u_{mk} | \nabla_k | u_{nk} \rangle}{(E_{nk} - E_{mk})^2}. \tag{12}$$

Here, $|u_{nk}\rangle$ are the periodic Bloch eigenstates of the Floquet Hamiltonian $E_{nk}$ are the corresponding eigenvalues, and $\nabla_k$ is the gradient with respect to the crystal momentum $k$. The calculated Chern number is then validated by assessing the Hall conductance.

## 4  Inducing high-Chern number QAH states by illumination

The quantum anomalous Hall effect was first observed experimentally by chromium-doping the $Bi_2Se_3$ material family [4]. Transitions to states with higher Chern numbers require an exchange field on both surfaces of a TI film larger than the bulk band gap, being 0.28 eV in $Bi_2Se_3$ [26]. However, that would require high doping levels, which may shift the Fermi energy and/or degrade the material structure [11, 21]. It is therefore unlikely that high-Chern number QAH states can be reached experimentally by magnetic doping. Instead, we here demonstrate that high Chern number states can be stabilized if TI properties are tailored using CPL. For small sample thickness, such that there is a hybridization gap, the physical properties of the system can be controlled by applying light to just the top surface of the TI. Otherwise, TI can be grown on a transparent substrate [50] with a goal of controlling the emergent properties by independently applying light to either top and/or bottom surface of the TI [56]. Such relative independence of controlling surface states on separate TI surfaces can also be desirable for applications in chirality-controlled topological transistors [57].

For both surfaces of the TI film exposed to the same CPL, our calculated threshold light intensity for phase transitions to higher Chern number states is shown in Fig. 3 for films with thicknesses $N_{QL} = 2 - 16$ (data for $N_{QL}$ up to 30 is available, but is not shown as it follows the same trend). The transition threshold values to a desired higher Chern number state may be fine-tuned by adjusting the wavelength, which adjusts the gap and the penetration depth of light. It should be noted that the necessary condition for the formation of high-Chern number QAH states is that the system is sufficiently thick to have no hybridization gap, and that the sum of the exchange fields due to internal magnetization and applied CPL is larger than the bulk gap.

However, as explained in Sec. 3, the intensity of applied light in the high-frequency range is limited by the stability of the topological band structure of the material [58]. As the maximum light intensity that can be applied to $Bi_2Se_3$ is $\tilde{A} \approx 4$ nm$^{-1}$, the high-Chern states in Fig. 3 will remain (borderline) unattainable when applying CPL to pristine $Bi_2Se_3$. Instead, we demonstrate that doping $Bi_2Se_3$ with magnetic atoms (as readily available experimentally [59]) circumvents this challenge and lowers the required intensity of light to induce the high-Chern

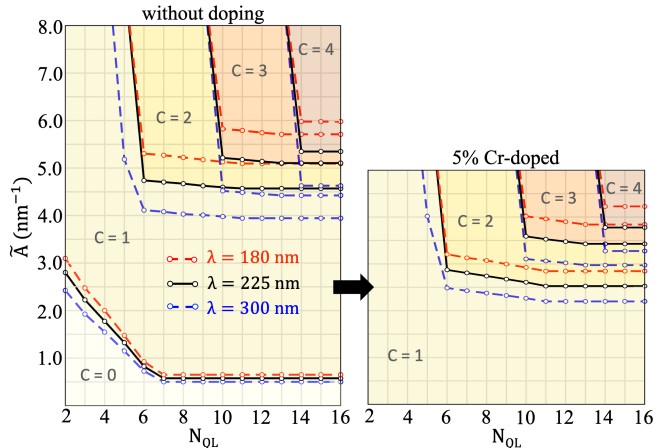

Figure 3: The phase diagram of high-Chern number states for thin-film $Bi_2Se_3$ exposed to CPL (of wavelength 180, 225 or 300 nm) on both surfaces, as a function of light intensity and sample thickness. By doping 5% Cr to $Bi_2Se_3$, achieving high-Chern number states with light intensity below the permitted threshold ($\approx 4$ nm$^{-1}$ for $Bi_2Se_3$) becomes feasible.

number QAH states. Adding magnetic impurities to TI decreases the overall SOC in the material, and increasing their concentration can change the material's topological phase. Here we have shown the results for a 5% concentration of chromium dopants since that is the highest experimentally accessible concentration without losing the key topological properties of the material. It is obvious that for lower concentrations, due to the reduction of the Zeeman exchange field, one requires higher intensity of light to achieve the phase transition, where the needed intensity of light will likely exceed the maximum allowed value that disturbs the topological properties of the material.

Specifically, we introduce 5% chromium doping (as experimentally realized in Ref. [60]), considering the magnetic impurities as inducing the exchange field in the Hamiltonian [11]. In the calculation, the magnetic impurity concentration is defined as $n_{imp} = (N_{imp}/N)$, where $N_{imp}$ dopants are randomly distributed over N lattice sites. Cr dopants in $Bi_2Se_3$ are considered dispersed across both Bi and Se lattice sites [60], and all have uniform size and orientation of their magnetic moments. For 5% Cr concentration, magnetic dopants create a 0.2 eV gap in the surface states, and the system resides in the QAH state (with $C = 1$). The resulting phase diagram under illumination is shown in Fig. 3, and demonstrates that under 5% Cr doping the high-Chern number states up to $C = 4$ (and larger for larger thickness) become reachable with permitted light intensity for $Bi_2Se_3$. This conclusively shows that the light intensity threshold for transition to $C > 1$ states is lowered to experimentally-applicable levels through dilute magnetic doping.

# 5 Axion states by illumination

Last but not least, we discuss the possibility of stabilizing axion insulator (AI) state in pristine $Bi_2Se_3$ films by suitable illumination. As a minimal requirement, the axion state warrants that the TI attains opposite gaps in top and bottom surface states (i.e. $\Delta_t \Delta_b < 0$, indices $t/b$ referring to top/bottom). For a sample grown on a transparent substrate, the surface states gap can be controlled independently using CPL. In this case, one expects the AI state to emerge once CPL of opposite chirality ($\eta$) is applied to the top and bottom surfaces of the sample (i.e. $\eta_t \eta_b < 0$).

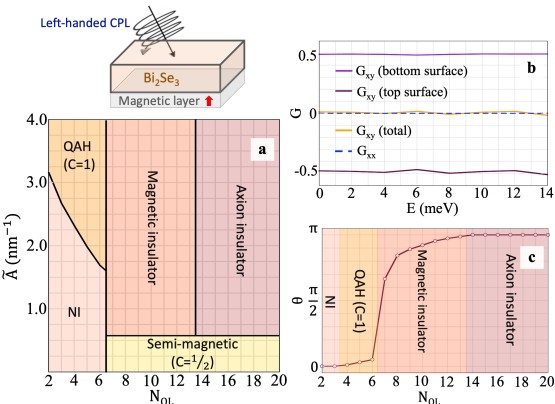

Figure 4: (a) The phase diagram for $Bi_2Se_3$ film on a magnetic layer (inducing exchange field $\Delta_b = 30$ meV at the interface, cf. above inset), with top surface exposed to left-handed CPL of wavelength $\lambda = 225$ nm, as a function of light intensity and sample thickness. (b) Hall conductance ($G_{xy}$) and longitudinal conductance ($G_{xx}$) of the 30QL thick sample, exposed to CPL of intensity $\tilde{A} = 2.5$ nm$^{-1}$, validating the axion insulator state. Panel (c) shows the calculated axion term ($\theta$) for samples of thickness 2-20QL under the same illumination. $\theta = \pi$ indicates that the system is in the AI phase for $N_{QL} \geq 14$.

To avoid the likely challenging requirement of two different light sources and a transparent substrate, we instead consider the TI film on top of a magnetic layer. We then apply CPL to regulate the gap in the top surface only, while the proximity to the magnetic layer induces the gap in the bottom surface. In that case, the Hall conductance in a four-probe setup can be used to identify the AI state, having Hall conductance $G_{xy} = \pm 1/2$ on opposite surfaces, while longitudinal $G_{xx}$ remains zero. We thus performed Landauer-Büttiker transport calculations for such a setup, that establishes the fundamental relation between the scattering amplitudes of a junction and its conducting properties in a non-interacting system without phase-breaking. The conductance at zero temperature is expressed using the Landau scattering method with the equation $G = \frac{e^2}{h} \sum_n T_n(E_F)$. For characterization of transport/conductance, we consider a four-terminal sample with leads 1 and 3 in the $x$-direction, and 2 and 4 in the $y$-direction. The transmission coefficient between leads $i$ and $j$ can be determined in terms of Green's functions as [61] $T_{ij} = \text{tr}[\Gamma_i G_{ij} \Gamma_j G_{ij}^\dagger]$, where $\Gamma_i$ describes the coupling of the device to the leads and can be expressed in terms of self-energy $\Sigma_i$ ($\Gamma_i = i[\Sigma_i - \Sigma_i^\dagger]$). Here $\Sigma_i$ can be considered as an effective Hamiltonian describing the lead-device interaction [62]. In this setup, four identical semi-infinite leads are attached to the sample, and the Hall conductance ($G_{xy}$) is given by [63] $G_{xy} = T_{14} - T_{12}$.

The obtained phase diagram as a function of the $Bi_2Se_3$ thickness and intensity of light applied to the top surface is shown in Fig. 4(a). Here we considered an exchange field of 30 meV at the bottom surface due to the adjacent magnetic layer, such as MnBi$_2$Se$_4$ [25]. The phase diagram is calculated for wavelength $\lambda = 225$ nm, and all phases are identified by their transport signatures. Specifically, for Fermi energies inside the gap, we encounter a normal insulator (NI) if $G_{xx} = G_{xy} = 0$, while if $G_{xx} = 0$ and $G_{xy} = 1$ the system is in the QAH phase. In the semi-magnetic phase, $G_{xx} = 1$ and $G_{xy} = \pm 1/2$ while Fermi energy is inside the proximity gap. For sufficiently thick samples, such that the hybridization gap vanishes, $G_{xx}$ will be zero as the light intensity increases. In this case, when $G_{xy}$ is half-quantized and opposite on opposite surfaces and the total Hall conductance is zero, we have an AI phase, and if $G_{xy}$ is not half-quantized and the total Hall conductance deviates from zero, the system is in the magnetic insulator phase.

We find axion insulator states stable for larger sample thickness ($N_{QL} > 13$) and moderate intensity of the light ($\tilde{A} > 0.575$ nm$^{-1}$). To validate that these states are indeed axion ones, Fig. 4(b) shows the calculated Hall conductance at the top and bottom surfaces of 30QL thick Bi$_2$Se$_3$ on a magnetic layer (as shown in the inset), for the top surface exposed to left-handed CPL ($\eta = -1$) of wavelength $\lambda = 225$ nm and intensity $\tilde{A} = 2.5$ nm$^{-1}$. The obtained half-quantized Hall conductances on the two surfaces and G$_{xx} = 0$ indeed prove the emergence of the AI state.

Additional validation of axion states in this structure is obtained by applying a probe electric field at described CPL exposure ($\eta = -1$, $\lambda = 225$ nm, and $\tilde{A} = 2.5$ nm$^{-1}$), and calculating the axion term $\theta$ [64, 65] as a function of thickness. For the infinite quasi two dimensional (2D) slab, we employ the expression of the ground state orbital magnetization in 2D [66, 67] as:

$$M_z(E_z) = -\frac{e}{\hbar} \frac{1}{(2\pi)^2} Im \sum_{n,n'} \int \left[ \frac{(E_{nk} + E_{n'k} - 2E_F)}{(E_{nk} - E_{n'k})^2} \langle u_{nk}| v_y(k) |u_{n'k}\rangle \langle u_{n'k}| v_x(k) |u_{nk}\rangle \, d^2k \right], \quad (13)$$

where u$_{nk}$ ($n/n'$ for occupied/unoccupied bands) is the periodic part of the Bloch wave function, E$_{nk}$ is its corresponding band energy. Calculations were done for a set of electric field strengths in the regime where the dependence of magnetization on electric field is linear. It is worth noting that in Floquet systems, the idea of a ground state loses its conventional meaning, which is typically used to describe an equilibrium state. Therefore, just like the computations related to the Chern invariant, we incorporate the impact of applied light to the onsite energies of the Hamiltonian. Then we use Eq. 13 to calculate the orbital magnetization. When the system is in the NI phase, $\theta = 0$, and in the QAH phase, the axion term has values close to zero. $\theta$ may have any value between 0 and $\pi$ in the magnetic insulator phase ($\theta \propto \Delta$) [64]. As plotted in Fig. 4(c), $\theta = \pi$ at all TI thicknesses beyond 13QL, so the axion state is indeed stabilized by illumination for the exemplified amplitude and wavelength of light.

With above validation, we finally summarize the phase diagram shown in Fig. 4. In absence of illumination, in the ultra-thin film regime, such that the wave functions of the surface states overlap, the presence of the magnetic layer breaks the TRS and positions the system in the NI phase (for $N_{QL} < 7$). By increasing the light intensity in this regime, if the induced gap is larger than the hybridization gap, the system enters the QAH phase with $C = 1$. The hybridization gap vanishes for larger sample thickness (thin-film regime), and weak applied light cannot induce a gap. In this case, $C = 1/2$, and the system is in the semi-magnetic phase (for $N_{QL} \geq 7$). As the light intensity is increased in the thin-film regime, the system enters either the magnetic insulator ($7 \leq N_{QL} < 14$) or the axion insulator phase ($N_{QL} \geq 14$), depending on the thickness. The former is encountered due to the interacting effects of magnetization on the bottom surface and the light on the top surface. For sample thickness exceeding the penetration depth of the light, the latter interaction ceases, and axion insulator state is firmly stabilized.

It is worth noting here that changing the chirality of applied CPL in this case enables topological switching between the magnetic insulator or AI phase (found for left-handed CPL and having $C = 0$) and the QAH state with $C = 1$ (for right-handed CPL and $\Delta_t \Delta_b > 0$).

## 6  Summary and Conclusions

To summarize, we have employed the real-space tight-binding effective model and Landauer-Büttiker transport calculations in combination with Floquet theory to reveal the possible topological phases and transitions in thin films of Bi$_2$Se$_3$, when exposed to circularly polarized light (CPL). In this quest, we have put particular attention to stable high-Chern number QAH and axion insulator phases, which are of particular recent interest for both their fundamental value and possible applications in emergent quantum devices. In their proposed realizations

to date, stabilization and identification of those phases has been prone to significant practical challenges. Our results instead advocate for the use of CPL to create and control desired states in a topological insulator, including QAH, semi-magnetic, magnetic, high-Chern number, and axion insulator states, in a convenient and fully externally controlled (contactless) fashion.

To be specific, we provided arguments that ultraviolet CPL of moderate intensity suffices to Floquet engineer multiple topological phases in $Bi_2Se_3$ films with thickness in the range 2-30 quintuple layers, and generated complete phase diagrams and parametric ranges for the stability of each relevant phase. To achieve that, we properly considered the penetration depth of light in the sample for the taken wavelength, and the corresponding variations of the gap due to such illumination.

For considered wavelength of illumination in the ultraviolet regime, we found that exposure to light on one surface can stabilize the QAH state for sample thickness up to about 15QL (exact threshold depending on exact wavelength), and semi-magnetic state for larger thickness. Realizing high-Chern number states requires exchange gap on both surfaces to exceed the bulk one, hence requires exposure to high-intensity light on both surfaces. We argue that one could indeed realize the latter setup by using a transparent substrate, but we also show that needed intensity of illumination would likely exceed the maximum permitted by the stability of the topological band structure. As a solution, we demonstrate that even a very dilute doping of $Bi_2Se_3$ by Cr would strongly reduce the required intensity of applied CPL, such that high-Chern number QAH states can be realized under realistic experimental conditions.

In case when CPL of opposite chirality is applied to opposite surfaces, one expects to realize the axion insulator state. As a more practical setup, we propose to have the TI film deposited on a magnetic layer, and expose just the top surface to the CPL, with chirality that induces a gap in the surface state opposite to the gap induced by magnetization at the bottom interface. In such a case, proper discrimination of possible topological phases requires transport calculations, as well as the calculation of the axion term. In doing so, we validated that axion insulator phase is indeed stable for moderately strong ultraviolet illumination, and sample thicknesses that exceed the penetration depth of the light. Thinner samples are instead in a magnetic insulator phase, or the QAH phase in the ultrathin regime (with hybridization gap present). Finally we point out that simply reversing the chirality of applied CPL provides a direct switch of axion or magnetic insulator phase into a QAH state, detectable in a four-probe transport setup.

Although the emergence of QAH and axion insulator states in TIs under CPL has been investigated recently [29, 32, 68], the realization of high-Chern number QAH phase in three-dimensional TIs belongs among the novelties of this work. In addition, our calculations within a real space model include important features that were not properly considered to date, such as the overlap of surface states wave functions, penetration depth of the light in the given material, the limitations of applied light intensity with respect to the stability of the electronic properties of the material under illumination, and we calculated the phase transition diagrams as a function of the sample thickness. All these parameters have a profound effect on the phase transition threshold values as well as the emergence of different phases. For example, it is not possible to have high-Chern number QAH states for samples with thickness below 6QL in the presence of a hybridization gap. Also, considering that it is crucial to fabricate this class of materials in a thin film regime to reduce bulk effects, it is important to understand different phases according to the thickness of the sample, which has indeed been carefully examined in our work. Moreover, we have shown that realization of high-Chern number QAH states by Floquet engineering is not feasible in pristine $Bi_2Se_3$ due to the need for an exchange field larger than the bulk gap, i.e. the need for light intensity beyond the one that would violate the characteristic electronic structure of the material; instead, we suggested the emergence of these states in a magnetically doped TI at experimentally accessible light intensity allowed for $Bi_2Se_3$.

All taken into account, our results pave the way to realize experimentally topological phases by demand in thin-films of topological insulators, externally controlled by tailored illumination. Besides the prospects for further fundamental studies of particular phases such as high-Chern number QAH or axion ones, the here demonstrated rich sensitivity of a well-established topological insulator $Bi_2Se_3$ to illumination promises to find its use in energy-efficient optoelectronics, advanced sensing, and other quantum technology.

## Acknowledgments

**Funding information** This research was supported by the Research Foundation-Flanders (FWO-Vlaanderen), the Special Research Funds (BOF) of the University of Antwerp, the Isfahan University of Technology, and the FWO-FNRS EoS-ShapeME project.

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
