# Peer review of "Floquet engineering of axion and high-Chern number phases in a topological insulator under illumination"

_SciPost Physics Core, doi:SciPost Phys. Core 7, 024 (2024)_

## Round 1 · Referee Report · Anonymous (Referee 1) · 2024-2-22

Strengths

See Report

Weaknesses

See Report

Report

This manuscript presents a theoretical study on the manipulation of topological phases in thin films of Bi2Se3 using circularly polarized light via Floquet engineering. It explores the potential for creating quantum anomalous Hall, high-Chern number, and axion insulator phases, each characterized by distinct Chern numbers. These phases are of great interest for their unique properties and potential applications in spintronics and optoelectronics. I support its publication, provided the authors address the following issues:

  1. Required: The key results of the paper are primarily derived from calculating Chern numbers for different thin film thicknesses, light wavelengths, and intensities. Importantly, the authors consider the fact that the driving field decays inside the sample. I believe that presenting the explicit expression (presumably, a formula that sums over the layer index, while in-plane wave vectors remain good quantum numbers) for calculating the Chern number with the layer-dependent vector potential, which decays exponentially inside the bulk, is crucial.

  2. Required: Similarly, it is crucial to present the explicit expression for calculating the orbital magnetization (M_z(E_z)), taking into account the decay of the driving field inside the sample. It is important to note that in Floquet systems, the concept of a "ground state" loses its traditional meaning - Equation (10) directly references Ref. [60], which is a study of an equilibrium state.

  3. Suggested: After the initial stage of identifying Floquet engineering as an interesting method for manipulating material properties, researchers gradually realized that the filling of the Floquet bands does not simply follow Fermi-Dirac statistics, and the so-called Floquet 'topological' states "may not give rise to quantized transport" [see Nature Reviews Physics 2, 229–244 (2020)]. To date, experimental transport observations have also failed to produce a quantized plateau [see Nature Physics 16, 38–41 (2020)]. Hence, it is recommended to highlight this aspect to raise awareness within the community. The topological classifications of Floquet systems are not adequately captured by the effective Hamiltonian.

  4. Suggested: Some minor suggestions:

4.1 The symbols $A_x$, $A_y$, $A_z$ in the tight-binding Hamiltonian and the periodic field $\boldsymbol{A}$ use the same notation. A change of notation is advised.

4.2 Calculating the maximum permitted power density of the laser illumination (e.g., W/cm$^2$) instead of just using $\tilde{A}$, would benefit the reader. And up to now, only transient Floquet states have been observed when driven by laser, as the laser rapidly heats the sample. It would be beneficial for the reader if this aspect were mentioned.

Requested changes

See Report

  • validity: -
  • significance: -
  • originality: -
  • clarity: -
  • formatting: -
  • grammar: -

Author:  Mohammad Shafiei  on 2024-02-27  [id 4328]

(in reply to Report 1 on 2024-02-22)
Category:
answer to question
reply to objection

I attached the response letter.

Attachment:

Floquet_scipost_pdf.pdf

Anonymous on 2024-03-05  [id 4335]

(in reply to Mohammad Shafiei on 2024-02-27 [id 4328])

I remain unsatisfied with the author's response, as it fails to address my primary concerns.

1) The author explained the calculation of the Chern number as: $C=2 \int_{B z} \hat{\boldsymbol{R}} \cdot\left(\frac{\partial \hat{\boldsymbol{R}}}{\partial k_x} \times \frac{\partial \hat{\boldsymbol{R}}}{\partial k_y}\right) \frac{d k_x d k_y}{4 \pi}$ with $\hat{\mathbf{R}}=\mathbf{R} / \mathrm{R}$ and $\mathbf{R}=(-v_F k_y, v_F k_x, M)$. However, the explanation still lacks clarity on how layer-dependent illumination affects $\mathbf{R}$.

2) Similarly, a clear explanation of how layer-dependent illumination is integrated into Equation 10 is required.

3) The revised manuscript has not been submitted for review.

---

## Round 1 · Referee Report · Anonymous (Referee 2) · 2024-4-3

Strengths

1. Systematic theoretical discussion, combining the Floquet formalism with the tight binding approximation of possible topological phases in Bi2Se3 and (Bi,Cr)2Se3 generated by circularly polarized light. High Chern numbers are predicted for strong intensities.

Weaknesses

1. Unsatisfactory discussion of experimental constraints
- decoherence by Coulomb interactions and spontaneous emissions (phonons, photons, plasmons, …)
- penetration length – mentioned but miscalculated, as only the intraband term is considered (Eq. 9), omitting a much more important interband contribution in the wavelength region of interest. If the linear response theory applies, the absorption coefficient (the inverse penetration length) is given by the imaginary and real parts of the dielectric function. Surprisingly, however, the magnitudes of the penetration depth for many semiconductors [e.g., J. Electronic Materials (2022) 51:6082–6107] are not very different from the values in Fig. 1(a).

2. No reference to relevant papers on similar investigations, e.g., : P. Molignini et al. SciPost Phys. Core 6, 059 (2023) “Probing Chern number by opacity and topological phase transition by a nonlocal Chern marker”; X. Wen et al., arXiv:2307.07116; Phys. Rev. B 109, 085148 (2024) “Photoinduced high-Chern-number quantum anomalous Hall effect from higher-order topological insulators.”

3. Introduction, 2nd sentence: “Due to strong spin-orbit coupling, incorporating magnetization to TIs can lead to otherwise unattainable magnetic topological states, such as quantum anomalous Hall (QAH) effect - also known as Chern insulator [3, 4], the high-Chern number QAH states [5], and axion insulator (AI) phase [6–8].” Given the observation of the QAHE in graphene multilayers [e.g., Z. Lu et al., Nature 626, 759 (2024) “Fractional quantum anomalous Hall effect in multilayer graphene], where the spin-orbit interaction is weak, that sentence may require a revision.

Report

I am writing my report as a supplement to the previous report. In general, the manuscript follows up on the team's earlier papers on a specific material system, Bi2Se3 (or Cr-doped) (Refs. 9, 10, 18), now adapting the elaborated methodology for the Floquet physics.

Considering the presence of earlier studies on the topic, I would recommend the publication in SciPost Phys. Core.

Requested changes

I would recommend the authors' reaction to the weak points enlisted above.

---

## Round 2 · Author Response

Please see the preprint file.

---

## Editorial Decision

published